# Micelle-in-Liposomes for Sustained Delivery of Anticancer Agents That Promote Potent TRAIL-Induced Cancer Cell Apoptosis

**DOI:** 10.3390/molecules26010157

**Published:** 2020-12-31

**Authors:** Zhenjiang Zhang, Sagar B. Patel, Michael R. King

**Affiliations:** Department of Biomedical Engineering, Vanderbilt University, Nashville, TN 37212, USA; zhenjiang.zhang@vanderbilt.edu (Z.Z.); sagar.patel@vanderbilt.edu (S.B.P.)

**Keywords:** TRAIL resistance, TRAIL sensitization, piperlongumine, micelle stability, drug delivery, complex nanomaterials, combination therapy

## Abstract

Tumor necrosis factor (TNF)-related apoptosis-inducing ligand (TRAIL) induces cancer cell-specific apoptosis and has garnered intense interest as a promising agent for cancer treatment. However, the development of TRAIL has been hampered in part because most human cancer cells are resistant to TRAIL. A few small molecules including natural compounds such as piperlongumine (PL) have been reported to sensitize cancer cells to TRAIL. We prepared a novel type of nanomaterial, micelle-in-liposomes (MILs) for solubilization and delivery of PL. PL-loaded MILs were used to sensitize cancer cells to TRAIL. As visualized by cryo-TEM, micelles were successfully loaded inside the aqueous core of liposomes. The MILs increased the water solubility of PL by ~20 fold. A sustained PL release from MILs in physiologically relevant buffer over 7 days was achieved, indicating that the liposomes prevented premature drug release from the micelles in the MILs. Also demonstrated is a potent synergistic apoptotic effect in cancer cells by PL MILs in conjunction with liposomal TRAIL. MILs provide a new formulation and delivery vehicle for hydrophobic anticancer agents, which can be used alone or in combination with TRAIL to promote cancer cell death.

## 1. Introduction

Selective killing of cancer cells has been long pursued in the development of effective cancer treatment, while achieving little to no side effects. The cancer-specific apoptotic potential of tumor necrosis factor (TNF)-related apoptosis-inducing ligand (TRAIL) has attracted great attention among biologists and oncologists, with some studies using recombinant human TRAIL (rhTRAIL) reaching clinical trials [1]. TRAIL belongs to the TNF cytokine superfamily that specifically induces apoptosis in a broad spectrum of human cancer cell lines while sparing most healthy cells [2]. Our lab has developed TRAIL-conjugated liposomes that have proven to be successful in preventing cancer metastases by targeting and neutralizing tumor cells in the circulation in multiple murine tumor models [3,4,5,6], and in ex vivo experiments using cancer patient blood samples [7,8,9].

Despite the potent tumor-specific properties of TRAIL against a broad range of cancer cells, some cell lines have been found to be resistant to the effects of TRAIL [10]. To increase the therapeutic effectiveness of the protein against resistant cancer cells, a few small molecules including some chemotherapy drugs such as paclitaxel and bortezomib, and natural compounds such as piperlongumine (PL) and curcumin were reported for the sensitization of cancer cells to TRAIL [11,12,13,14,15]. PL was identified in a high throughput screening as a drug candidate that induces apoptosis selectively in cancer cells but has little apoptotic effect on normal cells [16]. In our previous studies, we demonstrated that PL sensitized certain cancer cells to potent TRAIL-mediated apoptosis [15,17]. However, due to the low water solubility of PL, a solubilization method is needed to further PL’s application for TRAIL sensitization for future in vitro and in vivo studies.

Polymeric micelles are nanoscale core-shell structures formed via self-assembly of amphiphilic copolymer in aqueous medium [18]. Polymeric micelles have been pursued as a promising drug delivery vehicle due to their capacity to hold water insoluble drugs inside the hydrophobic core of micelles [19]. Numerous polymeric micelle systems have been reported to increase drug solubility. However, micelles are known to dissociate and release encapsulated drug prematurely following dilution in cell culture medium or administration in the blood circulation [20].

In this paper, we report our efforts in the solubilization and delivery of PL using a novel type of nanomaterial, micelle-in-liposomes (MILs). PL was first solubilized with polymeric micelles, which were then encapsulated into the aqueous core of liposomes (Figure 1). Arguably considered the most successful drug delivery vehicle in clinical applications, liposomes have been engineered to avoid reticuloendothelial system (RES) trapping and enjoy a blood circulation time of >20 h [21,22]. The aim of containing drug-loaded micelles inside of liposomes is to avoid micelle dissociation and premature drug release when they are diluted for use. We found potent apoptotic effects of PL-loaded MILs in combination with liposomal TRAIL in prostate cancer cells. MIL may be used as a general delivery vehicle for hydrophobic anticancer agents to promote TRAIL-induced cancer-specific apoptosis.

## 2. Results

### 2.1. Preparation and Characterization of MILs

In theory, micelles need to be sufficiently small for their passive encapsulation into unilamellar liposomes whose average size can be well controlled in the range of 50 to 200 nm [23]. We chose an amphiphilic polymer Poly(ethylene glycol) methyl ether-poly(D,L lactide) (PEG-PDLLA) with a molecular weight of 7000 kD, to prepare micelles with a solvent evaporation method. The as-prepared micelles were measured to be ~22 nm in size on average by dynamic light scattering (DLS), which measured the hydrodynamic size of the micelles that include the hydrophilic corona, and ~16 nm on average by TEM with negative staining, which measured only the hydrophobic core of the micelles (Figure 2A,C). To encapsulate the micelles into liposomes, a suspension of micelles was used to hydrate a lipid film that was prepared by evaporation of the organic solvent from the lipid solution. The mixture was extruded through a 200 nm polycarbonate filter to reduce the liposome size. The non-encapsulated micelles were successfully removed by size exclusion chromatography (SEC) based on the distinctive difference between the size of the micelles and liposomes as shown in Figure 1. The SEC purified mixture was found to be ~167 nm on average by DLS (Figure 2D). Cryo-TEM revealed that the mixture consisted of both empty liposomes and MILs at a ratio of ~3:1, i.e., micelles were successfully encapsulated into ~25% of all liposomes. Notably, some MILs contained more than one micelle in their aqueous core (Figure 2D). As quantified by HPLC, the PL content in MILs reached 0.52 mg/mL, ~20 times higher than the solubility of PL in water [24].

### 2.2. Piperlongumine Release from PL MILs

To compare their drug-release performance, PL micelles and PL MILs were placed in dialysis bags against a large amount of PBS at 37 °C. The PL concentration in the buffer was measured over time by HPLC. As shown in the release profile, nearly 100% of PL was released from the formulation of PL micelles within the first 6 h, while PL MILs exhibited a sustained release for >1 week (Figure 3). These results support the idea that, in the MIL formulation, the liposomes prevented rapid drug release from the PL-loaded micelles.

### 2.3. Combination Cancer Cell Treatment with PL MILs and Liposomal TRAIL

PL has been reported to induce apoptosis specifically in cancer cells by downregulating several anti-apoptotic proteins that are known to cause resistance in cancer cells to TRAIL-mediated apoptosis [24,25]. Our lab previously discovered that PL synergized with TRAIL to stimulate potent apoptosis in several cancer cells, suggesting the combination of PL and TRAIL as a novel paradigm for the treatment of cancer [15,17]. In the present study, we used PC3 and DU145 cells which exhibit medium and high sensitivity to TRAIL, respectively, to test the cytotoxic effects of PL MILs in combination with liposomal TRAIL. In both of the cell lines, the PL micelles showed equivalent cytotoxic effects to free PL after 24 h of treatment (Figure 4). On the one hand, sustained PL release from PL MILs may result in a lower effective concentration in the cell culture medium during treatment, but on the other hand, MILs could have increased cellular uptake of PL [26]. The combination treatment of PL in DMSO or PL MILs at 15 µM and TRAIL liposomes exhibited a significantly higher apoptosis rate than each individual therapy alone (Figure 4).

## 3. Discussion

In drug discovery, poor water solubility of drug candidates leads to low bioavailability and has long been a major reason for the failure of drug candidates [27]. Some approved drugs with low water solubility suffer from low permeability, rapid metabolism and clearance from the body. New solubilization technologies are needed for these existing drugs and for the majority of the molecules in the discovery pipeline to improve their efficacy, safety and patient compliance. As a promising drug carrier, polymeric micelles have been extensively pursued, owing to their ability to solubilize hydrophobic drugs within their hydrophobic core [19]. However, their clinical translation has been hampered by low in vivo stability due to polymer dissociation upon dilution in blood following systematic administration, and clearance by the immune system [28,29]. Strategies being explored to increase micellar stability with the goal of improving drug delivery mostly fall into two categories: covalent crosslinking to form covalent bonds in micelles, and non-covalent crosslinking to enhance the intra-micelle interaction via electrostatic interaction and hydrogen bonding [30,31]. However, these approaches may increase complexity in the large-scale manufacturing of micelles. Moreover, they may not prevent micelle molecules from binding to serum proteins—another major reason for micelle dissociation in the blood circulation [20,32].

In this research, we encapsulated PL-loaded micelles into liposomes as visualized by cryo-TEM. Sustained drug release from the complex MIL structure was achieved in contrast to rapid release from free PL micelles. The MILs also increased the solubility of PL, potentially allowing the use of higher concentrations for both in vitro and in vivo applications. Since micelles have been used to solubilize numerous hydrophobic small molecules, MIL may be further explored as a general delivery vehicle for water-insoluble drugs.

While this current study was conducted, other researchers have reported efforts to encapsulate micelles into liposomes for the delivery of different hydrophobic compounds. Zhang et al. and Franzè et al. both demonstrated a more sustained drug release from MIL than from micelles as we found in our research. However, neither of these two other papers provided a cryo-TEM image or other direct evidence to verify that the micelles were actually inside the aqueous core of the liposomes [33,34]. While Romana et al. did provide a cryo-TEM image of a single liposome containing worm-like micelles inside, no ratio of liposomes containing micelles over all liposomes was reported [35].

Liposomes have been explored for the encapsulation of macromolecules such as DNA, RNA and protein for delivery, in addition to their more popular use in the delivery of small molecules [36,37]. However, the loading efficiency of macromolecules into the aqueous core of liposomes has been less satisfactory, due to the large size of macromolecules and the slow diffusion rate during the encapsulation process [38,39,40]. In our preparation of MILs, as revealed by cryo-TEM, ~25% of all liposomes trapped micelles inside via passive drug loading. This encapsulation efficiency is lower than that for small drug loading where almost all the liposomes encapsulate a useful amount of drug. To further increase the encapsulation efficiency, electrostatic attraction, hydrogen bonding and even covalent bonding between micelles and liposomes may be explored [38].

## 4. Materials and Methods

### 4.1. Key Materials

The amphiphilic polymer PEG-PDLLA (PEG average Mn 2000, PDLLA average Mn 5000) used to prepare polymeric micelles was obtained from Sigma-Aldrich (St. Louis, MO, USA). Lipids for liposome preparation including Egg phosphatidylcholine (PC), cholesterol and 1,2-distearoyl-sn-glycero-3-phosphoethanolamine-N-[(polyethylene glycol)-2000] (DSPE-PEG) was obtained from Avanti Polar Lipids (Alabaster, AL, USA). Recombinant human TRAIL was purchased from Peprotech (Rocky Hill, NJ, USA). Liposomal TRAIL was prepared following our previously published methods [4,5]. Human prostate cancer PC3 and DU145 cells were obtained from American Type Culture Collection (ATCC) (Rockville, MD, USA) and were cultured in F-12k and EMEM medium, respectively, with 10% FBS.

### 4.2. Preparation of Micelles

The PL micelles were made using a solvent evaporation method [41]. Briefly, 10 mg PEG-PDLLA and 2 mg PL was dissolved in 6 mL acetonitrile before adding 2 mL of deionized (DI) water. Acetonitrile was mostly removed on a rotary evaporator. The resulting suspension was then dialyzed against 2 L DI water saturated with PL overnight to remove the organic solvent completely. The resulting mixture was filtered through a 0.22 µm syringe filter before being analyzed by ZetaSizer and TEM.

### 4.3. Preparation of Micelle-in-Liposomes (MILs)

MILs were prepared using a lipid film hydration method that is widely used to prepare liposomes [42]. Instead of using a buffer, a highly concentrated micelle suspension was used to hydrate the lipid film prepared with a molar composition of Egg PC:Cholesterol:DSPE-PEG = 64%:32%:4%. The hydration was allowed to proceed for one hour at room temperature before the mixture was extruded through a 200 nm polycarbonate membrane 10 times. The resulting MILs were then purified with SEC integrated with fast protein liquid chromatography (FPLC) to remove the non-encapsulated micelles. PL content in the purified MILs was quantified by HPLC using gradient conditions: acetonitrile:water:trifluoroacetic acid (20:80:0.01) to (80:20:0.01) over 20 min. A standard curve of PL at known concentrations was established for the measurement. Size distribution and morphology of the MILs was analyzed by Zetasizer and cryo-TEM.

### 4.4. Drug Release Test

One milliliter of purified PL micelles or PL MILs was placed in a dialysis bag with molecular weight cut off (MWCO) of 3.5 kD against 200 mL of pH 7.0 PBS buffer containing 0.5% Tween 20 at 37 °C. Released PL in the buffer was sampled over time and mixed with acetonitrile before being analyzed with HPLC using the HPLC conditions described above.

### 4.5. Cell Apoptosis Assay

Human prostate cancer cells PC3 and DU145 were seeded in 12-well plates at an initial density of 10^5^ cells per well, 24 h before treatment. Media was changed immediately before treatment. Cells were incubated with PL and PL MILs at 5 or 15 µM, or liposomal TRAIL at 100 ng/mL for PC3 cells and 200 ng/mL for DU145 cells, or both MILs and liposomal TRAIL at these concentrations, for 24 h before analysis by Annexin V/Propidium Iodide (PI) apoptosis assay using a Guava^®^ easyCyte™ 11HT flow cytometer (MilliporeSigma, Burlington, MA, USA) to quantify the proportion of viable cells.

## 5. Conclusions

We have developed a novel type of complex nanomaterial, MIL, and demonstrated its potential to formulate and deliver hydrophobic anticancer agents using PL as a model drug. Potent apoptotic effects of PL MILs in combination with liposomal TRAIL were observed in prostate cancer cells. The MILs provide a new approach for the delivery of water insoluble anticancer agents, which can be used either alone or in combination with TRAIL to promote its cancer-specific apoptotic effects. As MILs exhibited a much more sustained drug-release property than micelles in physiologically relevant buffers, MILs hold great promise to prolong the plasma life of micelles in vivo by containing the micelles inside liposomes before reaching their target.

## Figures and Tables

**Figure 1 molecules-26-00157-f001:**
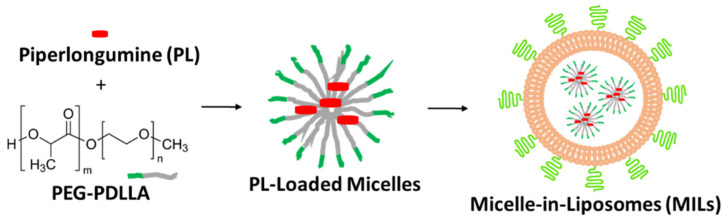
Schematic illustration of the formulation process of micelle-in-liposomes (MILs).

**Figure 2 molecules-26-00157-f002:**
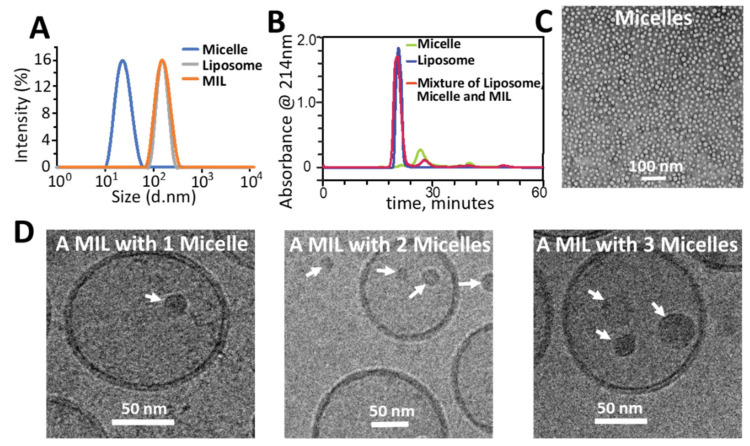
Physical characterization and purification of micelle-in-liposomes (MILs). (**A**) Size distribution of micelles, liposomes and MILs. (**B**) Separation of MILs from non-encapsulated micelles by size exclusion chromatography (SEC). (**C**) TEM of micelles. (**D**) Cryo-TEM of MIL containing 1, 2 or 3 micelles.

**Figure 3 molecules-26-00157-f003:**
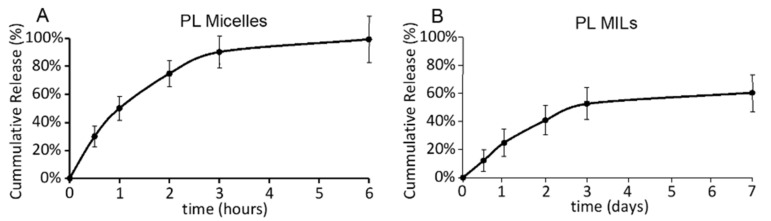
PL release profile from piperlongumine (PL) micelles (**A**) and PL MILs (**B**). PL micelles and PL MILs were placed in dialysis bags against a large amount buffer at 37 °C. PL released in the buffer was quantified by HPLC. Results are presented as the mean ± SEM, n = 3.

**Figure 4 molecules-26-00157-f004:**
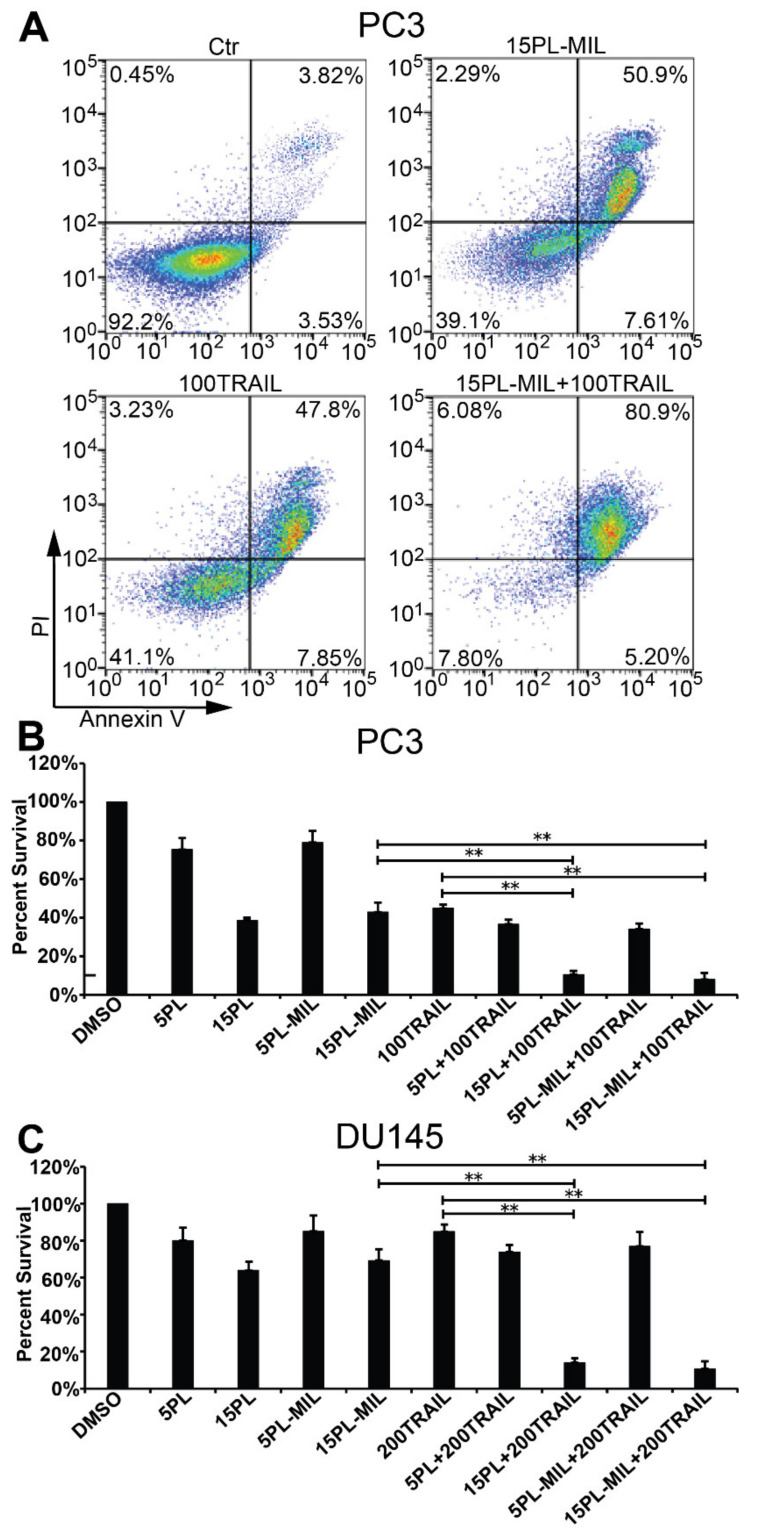
The anti-tumor effect of PL encapsulated MIL (PL-MIL) and liposomal tumor necrosis factor (TNF)-related apoptosis-inducing ligand (TRAIL). Prostate cancer cells PC3 (**A**,**B**) and DU145 (**C**) were treated with the indicated concentrations of PL in DMSO or PL-MIL (5 or 15 μM) and/or liposomal TRAIL (100 or 200 ng/mL) in 12-well plates for 24 h. The degree of cell apoptosis was measured using Annexin V/PI assay with flow cytometry. Representative flow cytometry dot plots from a PC3 experimental condition are shown.

## Data Availability

The data that support the findings of this study are available from the authors upon request.

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
