# Peer review of "Micelle-in-Liposomes for Sustained Delivery of Anticancer Agents That Promote Potent TRAIL-Induced Cancer Cell Apoptosis"

_molecules, 2020, doi:10.3390/molecules26010157_

Round 1
Reviewer 1 Report
Overall the manuscript is well written, and the data presented support the initial hypothesis that PL-MIL nanoformulation will increase the solubility of the hydrophobic drug (PL) and will increase its anticancer drug efficacy in combination with TRAIL. It will add to our repertoire of drug formulation for the delivery of other highly hydrophobic drugs.
My only concern is that I think a control is missing from fig 4:
Please show the comparison of PL+TRAIL versus PL-MIL +TRAIL for both the PC3 and DU145 cell lines in figure 4.
Author Response
Reviewer 1:
Overall the manuscript is well written, and the data presented support the initial hypothesis that PL-MIL nanoformulation will increase the solubility of the hydrophobic drug (PL) and will increase its anticancer drug efficacy in combination with TRAIL. It will add to our repertoire of drug formulation for the delivery of other highly hydrophobic drugs.
My only concern is that I think a control is missing from fig 4:
Please show the comparison of PL+TRAIL versus PL-MIL +TRAIL for both the PC3 and DU145 cell lines in figure 4.
Response: We appreciate the positive comments and have added PL+TRAIL, which showed similar results to PL-MIL+TRAIL, in Figure 4 as suggested.
Reviewer 2 Report
As it is presented, this paper has several problems, which are described below.
- In general, I could not find the innovation of this research. A similar concept was already published in the European Journal of Pharmaceutical Sciences in 2019 (doi: 10.1016/j.ejps.2019.01.013)
- Line 116: “Cell viability was measured using Annexin V/PI assay”; it is unacceptable to use Annexin V staining/PI as a synonym of cell viability assay. What is even more important, how the authors performed the Annexin V/PI staining? In general, this type of assay used a FACS analysis to determine the apoptotic population, late apoptotic (Annexin V positive and PI negative and necrotic population (Annexin V negative and PI positive population). For the first time, I see that researchers present data from Annexin V and PI double staining using “percent of survival.” Thus, I completely do not understand how the authors concluded that Pl combined with TRAIL might increase apoptosis. It is a very significant methodological flaw. Of course, PI alone is a good indicator of cell viability; however, if the authors want to determine the cytotoxic or antiproliferative effect, they should use a different assay. Moreover, it will be beneficial to compare the IC50 of tested systems against cancer cell lines. By the way, the cytotoxic effect should also be tested using a healthy cell line.
- Why did the authors not discuss the different sensitivity of PL-MIL against PC3 and DU145 cell lines? It is well-known that piperlongumine may affect the function of mutant p53. A study performed by Basak et al. showed that oxidative stress induced by PL might lead to functional restoration of mutant p53 and increase the cytotoxic effect of chemotherapeutic drugs such as BCNU doxorubicin. It is interesting because, in PC3, the p53 expression is lost due to a single base deletion at codon 138, while the DU145 cell line harbors a mutant of p53 (two-point mutations Phe223Leu and Val274Phe). It will be interesting to discuss also these findings.
- Could the authors clarify how they calculate the concentration of piperlongumine in MILs? The authors stated that >25% of all liposomes trapped micelles inside via passive drug loading. In general, this information means nothing because over 25% could be 50, 70, or 100%. This point should be clarified.
- In general, this study is not well constructed. The results did not support the conclusion; In Figure 4 (data from apoptosis detection or cell viability; I do not know which parameter was evaluated), if the authors want to show the potential of PL-MIL; they also should test the liposomal formulation of piperlongumine and also show data for PL in micelles only. Moreover, the “free piperlongumine” means PL dissolved in DMSO?
- In figure 3, n=3; it means three experiments or only triplicates from one experiment?
Author Response
Reviewer 2:
As it is presented, this paper has several problems, which are described below.
- In general, I could not find the innovation of this research. A similar concept was already published in the European Journal of Pharmaceutical Sciences in 2019 (doi: 10.1016/j.ejps.2019.01.013)
Response: We respectfully disagree with this reviewer’s assessment of the lack of innovation of our study. We have added a new paragraph in the Discussion to compare our research with the paper the reviewer refers to and two other papers mentioned by Reviewer 3.Our study represents one of the earliest to explore the possibility of encapsulating micelles into liposomes for hydrophobic drug delivery. Our study is also one of only two to provide cryo-TEM images to support that micelles can be successfully encapsulated into the aqueous core of liposomes. Ours is also the first study to report the ratio of liposomes containing micelles over all liposomes, which we believe is critical to evaluate the potential of this new delivery strategy for hydrophobic drugs.
- Line 116: “Cell viability was measured using Annexin V/PI assay”; it is unacceptable to use Annexin V staining/PI as a synonym of cell viability assay. What is even more important, how the authors performed the Annexin V/PI staining? In general, this type of assay used a FACS analysis to determine the apoptotic population, late apoptotic (Annexin V positive and PI negative and necrotic population (Annexin V negative and PI positive population). For the first time, I see that researchers present data from Annexin V and PI double staining using “percent of survival.” Thus, I completely do not understand how the authors concluded that Pl combined with TRAIL might increase apoptosis. It is a very significant methodological flaw. Of course, PI alone is a good indicator of cell viability; however, if the authors want to determine the cytotoxic or antiproliferative effect, they should use a different assay. Moreover, it will be beneficial to compare the IC50 of tested systems against cancer cell lines. By the way, the cytotoxic effect should also be tested using a healthy cell line.
Response: Annexin V in conjunction with Propidium iodide (PI) is widely used to determine if cells are viable, apoptotic, or necrotic through differences in plasma membrane integrity and permeability [Rieter et al., J Vis Exp. 2011; (50): 2597]. It is especially appropriate when used in the context of TRAIL delivery, which is well understood to act through the mechanism of apoptosis. This method has been used in dozens of publications by our group, including: Mitchell MJ et al., PNAS 111:930-5 (2014); Li J et al., Sci. Rep. 5:9987 (2015); Jyotsana N et al., Sci. Adv. 5:eaaw4197 (2019); Hope JM et al., Cell Death Dis. 10:837 (2019) and so on.
- Why did the authors not discuss the different sensitivity of PL-MIL against PC3 and DU145 cell lines? It is well-known that piperlongumine may affect the function of mutant p53. A study performed by Basak et al. showed that oxidative stress induced by PL might lead to functional restoration of mutant p53 and increase the cytotoxic effect of chemotherapeutic drugs such as BCNU doxorubicin. It is interesting because, in PC3, the p53 expression is lost due to a single base deletion at codon 138, while the DU145 cell line harbors a mutant of p53 (two-point mutations Phe223Leu and Val274Phe). It will be interesting to discuss also these findings.
Response: While we agree that the comparison between PC3 and DU145 might be an interesting study, it is beyond the scope of the current work which is aimed on reporting a new type of nanomaterial, micelles-in-liposomes (MILs), to formulate hydrophobic drugs. MILs may be explored as a general delivery vehicle for hydrophobic drugs. The cell experiments in our manuscript are intended to represent one example of how MILs can be used.
- Could the authors clarify how they calculate the concentration of piperlongumine in MILs? The authors stated that >25% of all liposomes trapped micelles inside via passive drug loading. In general, this information means nothing because over 25% could be 50, 70, or 100%. This point should be clarified.
Response: We appreciate the helpful comments and have change “>25%” to “~25%” meaning “about 25%” in the revised manuscript.
- In general, this study is not well constructed. The results did not support the conclusion; In Figure 4 (data from apoptosis detection or cell viability; I do not know which parameter was evaluated), if the authors want to show the potential of PL-MIL; they also should test the liposomal formulation of piperlongumine and also show data for PL in micelles only. Moreover, the “free piperlongumine” means PL dissolved in DMSO?
Response: In this manuscript, we report on a novel type of nanomaterial MILs that combine the advantages of both micelles and liposomes for the delivery of hydrophobic drugs, and we demonstrated a synergistic apoptotic effect in cancer cells from PL MILs in conjunction with liposomal TRAIL. We believe we have provided sufficient results to support the conclusions as agreed on by the three other reviewers. The Annexin V and PI negative populations are designated as viable/healthy cells, as generally done. We have added PL (in DMSO) + TRAIL in Figure 4 as we mentioned above in the response to Reviewer 1.
- In figure 3, n=3; it means three experiments or only triplicates from one experiment?
Response: n=3 refers to triplicates from one experiment, following general convention.
Reviewer 3 Report
Regarding to Ms.: molecules-1004723
Micelle-in-Liposomes for Sustained Delivery of Anticancer Agents that Promote Potent TRAIL Induced Cancer Cell Apoptosis
Zhenjiang Zhang, Sagar Patel and Michael R. King
To Authors,
In this study, the authors prepared and characterized polymeric micelles loaded with piperlongumine PL, which were included in liposomes. The new type of drug carrier, alone or in synergism with TRAIL (tumor necrosis factor-related apoptosis-inducing ligand (TRAIL) anti-cancer therapy), assured a improved water solubility for drug (x 20), a controlled release of drug from carrier (until 7 days) in simulated physiological conditions (PBS medium at 37 0C), and a rapid cytotoxicity (in maximum 24 h) in case of tumor cell lines tested in vitro.
The structures like micelles-contained liposomes represent a relative new trend in drug encapsulation/loading, applied to hydrophobic drugs. Thus, structures like micelles-contained liposome were prepared by other authors, too. Please, see:
- Drug Delivery, 2018,. 25(1), 611–622; Multiseed liposomal drug delivery system using micelle gradient as driving force to improve amphiphilic drug retention and its anti-tumor efficacy. Indeed they worked with another antitumor drug - amphiphilic asulacrine.
- European Journal of Pharmaceutics and Biopharmaceutics, Volume 154, September 2020, Pages 338-347; A liposome-micelle-hybrid (LMH) oral delivery system for poorly water-soluble drugs: Enhancing solubilisation and intestinal transport; Here it is about lovostatin, but same encapsulation technique of drug: micelles encapsulated in liposomes.
Compared with the old paper of authors (Ref. 17: Sharkey, C.C., et al., 2016), the novelty of this study (Manuscript ID: molecules-1004723) consists in the followings:
► use of another polymeric matrix which include drug PL, namely PEG-PLGA;
► preparation of another type of polymeric support for drug: micelles instead of nanoparticles;
► use of slightly different proportions and types of lipids for liposomes preparation;
►use of piperlongumine (PL) with/without TRAIL (tumor necrosis factor-related apoptosis-inducing ligand) as chemotherapeutic scheme.
Author Response
Reviewer 3:
To Authors,
In this study, the authors prepared and characterized polymeric micelles loaded with piperlongumine PL, which were included in liposomes. The new type of drug carrier, alone or in synergism with TRAIL (tumor necrosis factor-related apoptosis-inducing ligand (TRAIL) anti-cancer therapy), assured a improved water solubility for drug (x 20), a controlled release of drug from carrier (until 7 days) in simulated physiological conditions (PBS medium at 37 0C), and a rapid cytotoxicity (in maximum 24 h) in case of tumor cell lines tested in vitro.
The structures like micelles-contained liposomes represent a relative new trend in drug encapsulation/loading, applied to hydrophobic drugs. Thus, structures like micelles-contained liposome were prepared by other authors, too. Please, see:
- Drug Delivery, 2018,. 25(1), 611–622; Multiseed liposomal drug delivery system using micelle gradient as driving force to improve amphiphilic drug retention and its anti-tumor efficacy. Indeed they worked with another antitumor drug - amphiphilic asulacrine.
- European Journal of Pharmaceutics and Biopharmaceutics, Volume 154, September 2020, Pages 338-347; A liposome-micelle-hybrid (LMH) oral delivery system for poorly water-soluble drugs: Enhancing solubilisation and intestinal transport; Here it is about lovostatin, but same encapsulation technique of drug: micelles encapsulated in liposomes.
Compared with the old paper of authors (Ref. 17: Sharkey, C.C., et al., 2016), the novelty of this study (Manuscript ID: molecules-1004723) consists in the followings:
► use of another polymeric matrix which include drug PL, namely PEG-PLGA;
► preparation of another type of polymeric support for drug: micelles instead of nanoparticles;
► use of slightly different proportions and types of lipids for liposomes preparation;
►use of piperlongumine (PL) with/without TRAIL (tumor necrosis factor-related apoptosis-inducing ligand) as chemotherapeutic scheme.
Response: We appreciate the reviewer’s confirming comments of our results and conclusions in this manuscript. As mentioned in one of the responses to Reviewer 2, we have added a new paragraph in the Discussion to compare our research with the two papers the reviewer refers to as well as another paper mentioned by Reviewer 2. We appreciate that the reviewer has read one of and compared it to this manuscript. The major difference between the current work and our previously published paper [Sharkey, C.C., et al., TECHNOLOGY, 2016. 04(01): p. 60-69] is that, in this manuscript, we are reporting a new type of nanomaterial MILs that combine the advantages of both micelles and liposomes to prevent premature drug release and achieve a more sustained drug release than micelles alone.
Reviewer 4 Report
This manuscript presents good quality results of a speciation study focused on micelle-in-liposomes. The main research findings of this paper will be important for development of drug delivery. This is a well written, interesting, and useful contribution.
Author Response
Reviewer 4:
This manuscript presents good quality results of a speciation study focused on micelle-in-liposomes. The main research findings of this paper will be important for development of drug delivery. This is a well written, interesting, and useful contribution.
Response: We appreciate these positive comments.
Round 2
Reviewer 2 Report
I understand the use of Annexin-V and propidium iodide to determine the apoptotic induction. In cited papers:
10.1073/pnas.1316312111, in figure 1 the authors nicely presented the Propidium iodide/Annexin-V flow cytometry plots, which I could not find in the reviewed manuscript.
Li J et al., Sci. Rep. 5:9987 (2015), in figure 1 the authors presented cell viability evaluated using MTT, and in Figure 2 present in data from flow cytometry analysis (Annexin V/PI staining) for apoptosis/necrosis detection.
Writing my concern about the Annexin V/Propidium Iodide, I did not discredit the use of Annexin V/PI staining for apoptosis detection, but the way of data presentation, and as I wrote in my review reports, the terms “cell viability” and “apoptosis” are not synonyms. Of course, the Propidium Iodide positive cells are dead cells; thus, it gives an information about the percent of dead and live cells; however, in scientific paper, the terminology should be accurate. In the previous version, in Figure 4 caption, the authors stated that “Cell viability was measured using Annexin V/PI assay”; in the revised version, this information was changed to “The cell apoptosis was measured using Annexin V/PI assay” (however, the authors did not highlight this change); thus I suppose that the authors understand my concern.
Moreover, the authors should give information about the type of equipment used to do these experiments.
Author Response
Responses to Reviewers' Comments
Reviewer 2:
I understand the use of Annexin-V and propidium iodide to determine the apoptotic induction. In cited papers:
10.1073/pnas.1316312111, in figure 1 the authors nicely presented the Propidium iodide/Annexin-V flow cytometry plots, which I could not find in the reviewed manuscript.
Response: We agree with the reviewer that it would be more appropriate to show the original flow cytometry plots when possible. However, we have 10 groups of cells for each of the two types of cell line. Presenting all the plots would occupy too much space but doesn’t highlight the meaningful diffrences among groups. Therefore, we added a representative flow cytometry plot in Figure 4 and present the cell survival percentages of all groups as column figures to be concise and for clearer comparison among groups.
Li J et al., Sci. Rep. 5:9987 (2015), in figure 1 the authors presented cell viability evaluated using MTT, and in Figure 2 present in data from flow cytometry analysis (Annexin V/PI staining) for apoptosis/necrosis detection.
Response: We understand MTT assay measures the cell viability, and is actually easier to conduct than Annexin V/PI assay. We decided to change to use the latter to measure the apoptosis status of TRAIL-treated cells because TRAIL acts through apoptosis and the results may reflect the activity of TRAIL more accurately.
Writing my concern about the Annexin V/Propidium Iodide, I did not discredit the use of Annexin V/PI staining for apoptosis detection, but the way of data presentation, and as I wrote in my review reports, the terms “cell viability” and “apoptosis” are not synonyms. Of course, the Propidium Iodide positive cells are dead cells; thus, it gives an information about the percent of dead and live cells; however, in scientific paper, the terminology should be accurate. In the previous version, in Figure 4 caption, the authors stated that “Cell viability was measured using Annexin V/PI assay”; in the revised version, this information was changed to “The cell apoptosis was measured using Annexin V/PI assay” (however, the authors did not highlight this change); thus I suppose that the authors understand my concern.
Response: We appreciate that the reviewer pointed out the inaccurate use of terminology, and had revised our last manuscript accordingly.
Moreover, the authors should give information about the type of equipment used to do these experiments.
Response: We thank the reviewer for pointing out this omission, and have added the type of equipment in 4.5. Cell apoptosis assay.
